# Validity and Efficacy of the Elite HRV Smartphone Application during Slow-Paced Breathing

**DOI:** 10.3390/s23239496

**Published:** 2023-11-29

**Authors:** Joseph D. Vondrasek, Bryan L. Riemann, Gregory J. Grosicki, Andrew A. Flatt

**Affiliations:** Biodynamics and Human Performance Center, Department of Health Sciences and Kinesiology, Georgia Southern University (Armstrong), 11935 Abercorn St., Savannah, GA 31419, USA; briemann@georgiasouthern.edu (B.L.R.); ggrosicki@georgiasouthern.edu (G.J.G.); aflatt@georgiasouthern.edu (A.A.F.)

**Keywords:** heart rate variability, autonomic, parasympathetic, biofeedback, psychophysiology

## Abstract

Slow-paced breathing is a clinical intervention used to increase heart rate variability (HRV). The practice is made more accessible via cost-free smartphone applications like Elite HRV. We investigated whether Elite HRV can accurately measure and augment HRV via its slow-paced breathing feature. Twenty young adults completed one counterbalanced cross-over protocol involving 10 min each of supine spontaneous (SPONT) and paced (PACED; 6 breaths·min^−1^) breathing while RR intervals were simultaneously recorded via a Polar H10 paired with Elite HRV and reference electrocardiography (ECG). Individual differences in HRV between devices were predominately skewed, reflecting a tendency for Elite HRV to underestimate ECG-derived values. Skewness was typically driven by a limited number of outliers as median bias values were ≤1.3 ms and relative agreement was ≥*very large* for time-domain parameters. Despite no significant bias and ≥*large* relative agreement for frequency-domain parameters, limits of agreement (LOAs) were excessively wide and tended to be wider during PACED for all HRV parameters. PACED significantly increased low-frequency power (LF) for Elite HRV and ECG, and between-condition differences showed *very large* relative agreement. Elite HRV-guided slow-paced breathing effectively increased LF values, but it demonstrated greater precision during SPONT and in computing time-domain HRV.

## 1. Introduction

The autonomic nervous system (ANS), composed of the sympathetic (SNS; “fight-or-flight”) and parasympathetic nervous systems (PNS; “rest-and-digest”), regulates involuntarily controlled physiological phenomena such as heart rate (HR), blood pressure, respiration, and digestion [1,2]. Increased PNS, or vagal activity, decreases HR and increases variability in the inter-beat intervals, i.e., increased heart rate variability (HRV). Higher HRV is associated with greater physiological flexibility, adaptability, and self-regulation [3] and superior health and lifestyle indicators [4]. Conversely, lower HRV is common in older adults [5] and is associated with smoking [6], physical inactivity [7], and diabetes [8]. Low HRV is also associated with increased hypertension, cardiovascular disease, and mortality risk [9,10,11,12,13,14]. Thus, practical cost-effective strategies to increase vagal-mediated HRV may help to reduce disease risk and increase lifespan [15,16,17].

HRV biofeedback (HRVB) is a clinical intervention used to increase vagally mediated HRV. During HRVB, the individual performs slow-paced breathing at a resonance frequency of ~4.5 to 7 breaths∙min^−1^ [18,19,20,21]. During the breath cycle, HR normally increases during inhalation and decreases during exhalation, i.e., respiratory sinus arrhythmia [21,22]. When breathing at a resonance frequency during HRVB, peak-to-trough differences in HR are maximized, which increases HRV [19,23,24,25]. Purposely altering breathing rate and depth to maintain resonance frequency during HRVB can be leveraged to improve PNS activity (HRV) and various other indicators of health [26]. For example, HRVB has been used to improve cognition while reducing perceived total stress [20], reduce symptoms of trait anxiety [27], lower blood pressure with concomitant reductions in inflammatory markers [28], and improve mood [29]. HRVB is a clinical technique, but widely available low-cost smartphone applications (apps) offering slow-paced breathing guidance and HRVB [30] could be used outside of the clinic for improving general health and well-being [31]. However, investigation into the validity of low-cost HRVB apps is necessary before they can be recommended for personal use.

Elite HRV is a cost-free smartphone app that offers daily HRV assessment [4,32] and, more recently, HRVB during slow-paced breathing. The app provides proprietary RR filtering and automatic HRV assessment after each reading, along with the option to export raw RR interval data for assessment in separate software. Multiple studies [33,34,35,36,37,38] have investigated the validity of Elite HRV, but few [35,36,38] have compared Elite HRV-derived HRV values to a gold standard (i.e., electrocardiography (ECG)), and have instead opted to compare HRV metrics generated from exported RR interval data. Several [33,34,35,36,37,38], but not all [39], studies investigating the accuracy of Elite HRV reported acceptable agreement, supporting the use of Elite HRV for measuring resting seated and supine HRV among healthy adults. However, of these investigations, none assessed the agreement between Elite HRV and a criterion during slow-paced breathing. Due to this gap in the literature, it is currently unknown whether Elite HRV can accurately measure the substantial increases in HRV expected to occur during slow-paced breathing. Therefore, we aimed to (1) assess the validity of Elite HRV for quantifying HRV parameters during spontaneous and slow-paced breathing and (2) verify the efficacy of Elite HRV-guided slow-paced breathing for increasing HRV. We hypothesized that (1) Elite HRV would provide acceptable agreement with ECG-derived HRV values during SPONT and PACED breathing and (2) Elite HRV-guided slow-paced breathing would increase HRV parameters.

## 2. Materials and Methods

### 2.1. Participants

We recruited a convenience sample of 22 healthy adults (13 M/9 F) from a college campus. The inclusion criterion was being aged between 18 and 39 years old. Exclusion criteria were the use of tobacco products; acute illness (flu, COVID-19, etc.); and/or reporting a known cardiovascular, metabolic, or neurological condition. Each participant provided written and informed consent before completing the study. All procedures were conducted in accordance with the Declaration of Helsinki and approved by the Institutional Review Board (Protocol #: H22076).

### 2.2. Study Design

This study used a single-visit counterbalanced cross-over design in which participants came to the laboratory between 0700 and 1000 after an overnight fast and having abstained from alcohol and structured exercise of any intensity for 24 h, caffeine for 12 h, and fluids for at least 1 h before their visit [40]. Agreement between Elite HRV and ECG-derived HRV was assessed via concurrent recordings in a supine position during 10 min of spontaneous (SPONT) and app-guided slow-paced (PACED) breathing. Ten min of supine stabilization preceded each condition, and 3 min of standing between conditions functioned as a washout period [41]. Agreement was assessed between ECG-derived HRV values and Elite HRV-derived values provided automatically by the app.

### 2.3. RR Interval Collection

After establishing written and informed consent, height and weight were measured and recorded. Participants were then directed to a dimly lit, temperature-controlled (21 °C) examination room for the experimental procedures. RR intervals were simultaneously recorded via ECG and Elite HRV using a Polar H10 (Polar Electro Oy, Kempele, Finland) during PACED and SPONT. The H10 was placed at the level of the xiphoid process and the strap was wetted for conductivity. The ECG electrodes were placed in a modified lead II configuration with data transmitted from the integration belt (Biopac BIONOMADIX, BIOPAC Systems Inc., Coletta, CA, USA) to the data acquisition system (Biopac MP 160, BIOPAC Systems Inc., Coletta, CA, USA). The sampling rate was set to 1000 Hz. AcqKnowledge software 5.0 (BIOPAC Systems Inc., Coletta, CA, USA) was used to integrate the ECG and respiratory belt data. For ECG analysis, raw AcqKnowledge files were exported into Kubios HRV Premium software (Version 4.2.1, University of Kuopio, Kuopio, Finland) [42]. ECG waveforms were manually inspected for abnormalities by two researchers (J.D.V., A.A.F.). Ectopic beats were de-selected as normal beats via the Kubios software creating an outlier inter-beat interval, which was subsequently removed with the lowest-threshold Kubios filter. The correction of only the identified abnormalities, and no other beats, was confirmed by visual inspection of the tachogram and by noting the number of corrected beats displayed via Kubios Premium software version 4.2.1. No RR interval detrending was applied to be consistent with Elite HRV methodology. The Polar H10 (1000 Hz) was connected via Bluetooth to Elite HRV (Version 5.5.1, Asheville, NC, USA) on an iPad (5th generation, Apple Inc., Cupertino, CA, USA). HRV results from Elite HRV displayed on the app were recorded for analysis. When Elite HRV recordings were started and stopped, a marker was placed on the ECG file in AcqKnowledge to match recording segments. Although Elite HRV performs automatic RR interval filtering, the algorithm is proprietary.

All HRV parameters were derived following standardized guidelines [1]. For time-domain HRV metrics, we compared the mean RR, the root mean square of successive differences (RMSSD), and the standard deviation of normal-to-normal RR intervals (SDNN) between Elite HRV and ECG during SPONT and PACED. For the frequency domain, we made the same comparisons for low-frequency (LF, 0.04–0.15 Hz) and high-frequency spectral power (HF, 0.15–0.4 Hz) [3]. Frequency-domain metrics were determined using the fast Fourier transformation based on Welch’s Periodogram method [43] to be consistent with Elite HRV. RMSSD and HF were considered markers of the PNS, LF a marker of baroreflex activity, and SDNN a marker of global variability [44].

### 2.4. Respiration Belt

A strain gauge respiration belt (BIOPAC SS5LB; BIOPAC Systems Inc., Coletta, CA, USA) was placed at the mid-point between the navel and xiphoid process. Respiration rate was transmitted from the data integration belt to the acquisition system for analysis in the AcqKnowledge software 5.0 to confirm compliance with app-guided slow-paced breathing and to quantify respiration rate differences between SPONT and PACED.

### 2.5. Breathing Conditions

PACED was guided by Elite HRV using the “Custom Breathing” program which involves continuous RR interval acquisition and subsequent automatic HRV calculation. The inhale and exhale duration were both set to 5 s (6 breaths·min^−1^) [41,45]. SPONT was assessed using the “Open HRV reading” program which also involves continuous RR interval acquisition and subsequent automatic HRV calculation. The iPad was positioned at a self-selected distance from the participant using a flexible gooseneck tablet holder (enGMOLPHY, Owosso, MI, USA) during both conditions. Before beginning PACED, participants were familiarized with the cues for inhaling and exhaling.

### 2.6. Statistical Analysis

Readings from the Elite HRV app are automatically rated as “good”, “okay”, or “poor” based on the quantity of artifacts corrected. If readings were rated as “poor”, the readings were excluded from the analysis. Prior to comparing the Elite HRV app to ECG, the normality of paired difference values was assessed via Shapiro–Wilks tests. Systematic bias testing was completed via paired *t*-tests or Wilcoxon Signed Rank tests [46]. The mean bias and 95% upper and lower (mean bias ± 1.96 × standard deviation of difference scores) limits of agreement (LOAs) were determined for comparisons that met the assumption of normally distributed difference scores. When this assumption was violated, the median bias and 95% range (2.5th, 97.5th percentiles) were used for LOA [47]. Heteroscedasticity was explored via visual inspection of Bland–Altman figures [48] and the degree of heteroscedasticity was assessed by calculating Kendall’s tau (τ) correlation between the absolute differences and corresponding means; heteroscedasticity was considered to be present when τ was >0.20 [49]. Relative agreement between ECG and Elite HRV was assessed via Lin’s concordance correlation coefficient (LCC) [50]. Agreement was further assessed using ordinary least products (OLP) regression [51]. In this analysis, the slope and intercept along with the 95% confidence interval (CI) for each were used to determine if there were proportional or fixed biases present. If the 95% CI for the slope and intercept did not contain 1 and 0, respectively, proportional and fixed biases were considered to be present [51].

To compare HRV parameters between conditions, the normality of paired difference values was assessed via Shapiro–Wilks tests. Paired *t*-tests or Wilcoxon Signed Rank tests were used to compare breathing rates and HRV parameters between conditions. LCC [50] was used to quantify the relative agreement between difference scores (i.e., SPONT vs. PACED). Relative agreement for all LCCs was qualitatively interpreted as <0.10 = trivial, <0.3 = small, <0.5 = moderate, <0.7 = large, <0.9 = very large, and >0.9 = near perfect [52]. The statistical significance threshold was set at α < 0.05. Statistical analysis was completed using JMP (Version 16; JMP, Cary, NC, USA), SPSS (Version 27, IBM Corp., Armon, NY, USA), and Excel (Version 16, Microsoft Corp., Redmond, WA, USA).

## 3. Results

Twenty-two people volunteered to participate in this study, but two were excluded. One participant was excluded because the signal quality was rated as “poor” by Elite HRV during PACED. The other participant was excluded because ECG signal quality was excessively noisy, making it difficult to classify beats as normal or abnormal. Thus, twenty participants (13 M/7 F; 23 ± 3.7 years; 23 ± 2.0 kg/m^2^) were included in the final analysis. The target breathing rate during PACED was achieved (6.1 ± 0.4 breaths∙min^−1^) and was significantly slower (*p* < 0.001) than SPONT (10.7 ± 1.3 breaths∙min^−1^).

Of the 40 total ECG recordings (20 participants x 2 conditions), 36 required no correction of ectopic beats or R-peak identification (Table 1).

### 3.1. Mean RR

Summary and agreement statistics for the between-device (ECG vs. Elite HRV) comparison of the mean RR are reported in Table 2 and Bland–Altman plots are displayed in Figure 1a,d. The mean RR differences between ECG and Elite HRV during SPONT were non-significant (*p* > 0.05), with *near-perfect* relative agreement and no evidence of fixed or proportional biases based on OLP results. However, there was evidence of heteroscedasticity based on τ. For PACED, a significant difference in mean RR was observed between ECG and Elite HRV (*p* < 0.05). However, there was *near-perfect* relative agreement and no evidence of fixed or proportional biases based on OLP results.

Summary statistics for the between-condition comparison (PACED vs. SPONT) of the mean RR are reported in Table 3. For both ECG and Elite HRV, the mean RR was significantly shorter during PACED compared to SPONT (*ps* < 0.05). Mean RR difference values for ECG and Elite HRV between conditions demonstrated *near-perfect* relative agreement (*p* < 0.05).

### 3.2. SDNN

Summary and agreement statistics for the between-device comparison of SDNN are reported in Table 2, and Bland–Altman plots are displayed in Figure 1b,e. SDNN differences between ECG and Elite HRV were significant during SPONT (*p* > 0.05) but not PACED (*p* > 0.05), and relative agreement was *near perfect* for each condition. Moreover, there was evidence of heteroscedasticity for SPONT and PACED SDNN based on τ, but there was no evidence of fixed or proportional biases based on OLP results.

Summary statistics for the between-condition comparison (PACED vs. SPONT) of SDNN are reported in Table 3. There was a significant increase in SDNN (*p* < 0.05) when measured by ECG, but not Elite HRV (*p* = 0.05). Between-condition SDNN difference values demonstrated *very large* relative agreement (*p* < 0.05).

### 3.3. RMSSD

Summary and agreement statistics for the between-device comparison of RMSSD are reported in Table 2, and Bland–Altman plots are displayed in Figure 1c,f. There were significant differences in RMSSD between ECG and Elite HRV during SPONT (*p* < 0.05), although relative agreement was *near perfect*. There was evidence of heteroscedasticity based on τ, and OLP results showed that Elite HRV tended to underestimate RMSSD. During PACED, there was also a significant difference (*p* > 0.05) observed between ECG and Elite HRV for RMSSD, with *very large* relative agreement. There was evidence of heteroscedasticity based on τ but no evidence of fixed or proportional biases based on OLP results.

Summary statistics for the between-condition comparison (PACED vs. SPONT) of RMSSD are reported in Table 3. When measured by ECG and Elite HRV, RMSSD was not different during PACED compared to SPONT (*p* > 0.05). The between-condition RMSSD difference values for ECG and Elite HRV demonstrated *large* relative agreement (*p* < 0.05).

### 3.4. LF

Summary and agreement statistics for the between-device comparison of LF are reported in Table 4, and Bland–Altman plots are displayed in Figure 2a,c. The LF differences between ECG and Elite HRV during SPONT and PACED were non-significant (*p* > 0.05), with *very large* to *near perfect* relative agreement. There was evidence of heteroscedasticity for SPONT and PACED LF based on τ, but no evidence of fixed or proportional biases based on OLP results. Notably for LF, LOAs were excessively wide due to large inter-individual variation in difference values.

Summary statistics for the between-condition comparison (PACED vs. SPONT) of LF are reported in Table 3. When measured by ECG and Elite HRV, LF was significantly greater during PACED (*p* < 0.05). Between-condition LF difference values demonstrated *very large* relative agreement (*p* < 0.05).

### 3.5. HF

Summary and agreement statistics for the between-device comparison of HF are reported in Table 4, and Bland–Altman plots are displayed in Figure 2b,d. The HF differences between ECG and Elite HRV during SPONT and PACED were non-significant (*p* < 0.05) with relative agreement rated as *near perfect* for SPONT and *moderate* for PACED. There was evidence of heteroscedasticity based on τ during SPONT and PACED, but no evidence of fixed or proportional biases based on OLP results. Notably for HF, LOAs were excessively wide due to large inter-individual variation in difference values.

Summary statistics for the between-condition comparison (PACED vs. SPONT) of HF are reported in Table 3. When measured by ECG and Elite HRV, HF was not different during PACED compared to SPONT (*p* > 0.05). The between-condition HF difference values demonstrated *moderate* relative agreement (*p* < 0.05).

## 4. Discussion

The purpose of this investigation was to (1) assess the validity of Elite HRV for quantifying HRV parameters during spontaneous (SPONT) and slow-paced (PACED) breathing and (2) verify the efficacy of Elite HRV-guided slow-paced breathing as a method to increase HRV. The main findings for our first aim were that Elite HRV demonstrated much stronger agreement with ECG for time-domain relative to frequency-domain parameters, as well as a tendency for stronger agreement during SPONT versus PACED. Regarding our second aim, Elite HRV-guided breathing effectively increased LF, as measured by ECG and Elite HRV, and individual changes in LF demonstrated *very large* relative agreement between measurement tools.

### 4.1. Comparison of ECG- and Elite HRV-Derived HRV

We observed excellent agreement between the ECG- and Elite HRV-derived mean RR during SPONT with median bias and LOA values comparable with parametric comparisons in previous investigations [35]. Some heteroscedasticity was noted such that difference scores tended to scatter with increasing mean RR (i.e., lower heart rate). Poorer agreement for mean RR during PACED, reflected in a significant median bias, was largely driven by two outlier data points, as 18/20 participants demonstrated individual difference values of ≤7 ms. RMSSD measured by Elite HRV was significantly different than ECG-derived values during SPONT and PACED, which agrees with work by Guzik et al. [39] (mean 4 ms underestimation of ECG-derived RMSSD; significant correlation coefficient of 0.67), but we did observe ≥*very large* relative agreement. Some heteroscedasticity was noted such that difference scores tended to scatter upwards with increasing RMSSD, which was more notable during PACED. Prior studies that reported excellent agreement for Elite HRV-derived RMSSD used the natural logarithm of RMSSD [35,36], which limits skewness, or they exported RR intervals for analysis (i.e., did not use Elite HRV-derived values) [34], which may help explain our discrepant findings. Despite *near-perfect* relative agreement for SDNN, we observed a significant difference between ECG and Elite HRV during SPONT, and heteroscedasticity was more strongly noted during PACED. However, similar to results for RMSSD, a limited number of outliers were driving sub-optimal agreement for SDNN, as demonstrated in the Bland–Altman plots. To our knowledge, only one previous study examined Elite HRV-derived SDNN and reported a significant mean bias (mean 5.2 ms underestimation of ECG-derived SDNN; significant correlation coefficient of 0.73) [39]. Interestingly, in all but one case in the current study (SDNN during SPONT), upper LOAs were substantially greater than the lower LOAs. This reflects the non-normality of difference scores and may indicate some level of consistency in Elite HRV erring towards the underestimation of ECG-derived parameters, most often when values were higher. Although parametric procedures were performed previously, visual inspections of Bland–Altman plots from Gurzik et al. suggest that similar non-normality in difference values were observed for RMSSD and SDNN [39].

Frequency-domain parameters (HF and LF) demonstrated excessively wide LOAs, especially during PACED, and heteroscedasticity was observed for each comparison. Relative agreement was ≥*large* for each comparison. We are aware of only one previous investigation that explored agreement for frequency-domain parameters [31], but LF and HF were computed after RR data were exported into Kubios (i.e., did not use Elite HRV-derived values). Despite using the same RR processing and HRV computation method, wide LOAs for LF and HF were reported, particularly when assessed in the standing position [34]. Mean bias values were also non-trivial (≥111.2 ms^2^) despite being non-significant (*p* > 0.05) [34]. In line with our findings of greater LOAs during PACED, Menghini et al. [53] compared PPG- and ECG-derived LF and HF and found that LOAs were wider during paced versus spontaneous breathing.

Potential sources of error between ECG- and Elite HRV-derived HRV using a Polar H10 have been previously described [54]. Discrepant values could be related to differences in ECG electrode placement versus Polar H10 chest strap location [55]. Differences in signal processing algorithms could also affect HRV results [56]. Additionally, HRV smartphone applications often use a threshold-based RR interval correction algorithm that corrects inter-beat intervals that exceed pre-determined values [57]. Thus, Elite HRV’s proprietary filtering method may be correcting long RR intervals of sinus node origin, which could help explain its tendency to underestimate ECG-derived results, particularly at higher HRV values, and during PACED. Finally, inter-individual differences in physical characteristics (e.g., cardiac dimensions, body composition), or underlying cardiac pathologies could also impact QRS complexes and the timing of inter-beat interval detection between measurement tools [54].

### 4.2. Comparison of SPONT and PACED HRV

The ECG- and Elite HRV-derived values for mean RR, SDNN, RMSSD, and HF provided comparable absolute differences between SPONT and PACED (Table 3). Additionally, there was ≥*large* relative agreement between difference scores for mean RR, SDNN, and RMSSD, but HF showed only *moderate* relative agreement. Our findings for mean RR, SDNN, RMSSD, and HF are in line with some, but not all, previous research on HRV responses to slow-paced breathing. For example, when comparing breathing at 6 breaths·min^−1^ with and without biofeedback, RMSSD and SDNN increased, HF increased at the group level with a wide range of error, and the mean RR increased [41]. We observed similar changes in SDNN (ECG-derived only), but RMSSD did not significantly change, and we noted a decrease in mean RR. Melo et al. [58] found that participants had higher RMSSD and SDNN during slow-paced breathing (6 breaths·min^−1^), which only matches our results for SDNN. A possible explanation for the reduction in mean RR and limited increases in RMSSD and HF in the current study could be that participants were more relaxed during SPONT versus PACED. Lack of familiarity with slow-paced breathing possibly kept them vigilant in adhering to the breathing cadence, resulting in stronger inhalations which contribute to HR accelerations and collectively may have limited the magnitude of decelerations.

LF was significantly increased irrespective of the HRV measurement tool, and difference scores demonstrated *very large* relative agreement. The observed increase in LF matches many previous investigations [41,58,59] and is pertinent to assessing the accuracy of Elite HRV during slow-paced breathing or HRVB. This is because LF is the primary HRV parameter of interest during breathing practices [25,60,61]. LF reliably increases during slow breathing in various populations [26] and is a key metric used to determine one’s resonance breathing frequency [25]. The increase in LF for both devices supports the utility of Elite HRV’s guided breathing feature for acutely increasing LF HRV.

### 4.3. Limitations

This study was not without limitations. The ECG recordings were marked at the moment that Elite HRV recordings started, but there was a possibility of very minor error in synchronizing the RR interval collection (e.g., <1 s). Participants completed this study in a quiet and temperature-controlled room, but the environment may not represent a comfortable at-home setting in which app-guided breathing would regularly be performed. Moreover, although participants were familiarized with the breathing cadence for PACED, they may not have been adequately familiarized with appropriate relaxation during slow-paced breathing which could have affected HRV responses to PACED versus SPONT. Sample size could also be considered another limitation. This study also had a few notable strengths. The testing procedures were in the morning after an overnight fast which created standardized conditions, and male and female participants were included in the study. Also, slow-paced breathing had not been previously used to validate Elite HRV or similar apps.

## 5. Conclusions

The current investigation revealed novel information pertaining to the validity and efficacy of the Elite HRV smartphone application. Elite HRV’s slow-paced breathing feature produced expected increases in LF HRV, supporting its efficacy as a cost-free option for transiently altering cardiac autonomic modulation. However, its agreement with reference ECG results varied between time- and frequency-domain parameters and between paced and spontaneous breathing conditions. Strong agreement was generally observed for time-domain HRV parameters with the exception of a small number of outliers that were observed most commonly at higher values. Thus, practitioners should be aware that Elite HRV may underestimate time-domain values in individuals with higher HRV. Contrastingly, although frequency-domain values tended to show acceptable relative agreement, individual bias values were generally large and heterogeneous, contributing to excessively wide agreement limits. Thus, caution should be used when examining Elite HRV-derived spectral HRV. Ongoing investigation is encouraged to keep up with application updates which can include modifications to RR filtering and HRV computation procedures and thus improve upon its agreement with reference ECG.

## Figures and Tables

**Figure 1 sensors-23-09496-f001:**
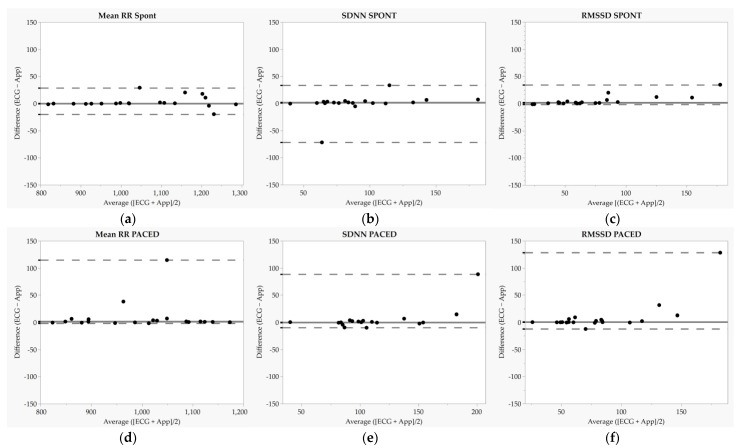
Bland–Altman plots for mean RR, standard deviation of normal-to-normal intervals (SDNN), and root mean square of successive differences (RMSSD). Each dot represents one participant. (**a**–**c**) represent Bland-Altman plots during the spontaneous breathing condition while (**d**–**f**) represent Bland-Altman plots during the paced breathing (6 breaths·min^−1^) condition. SPONT = spontaneous breathing; PACED = paced breathing (6 breaths·min^−1^); solid lines represent median biases, and dashed lines represent 2.5th and 97.5th percentiles.

**Figure 2 sensors-23-09496-f002:**
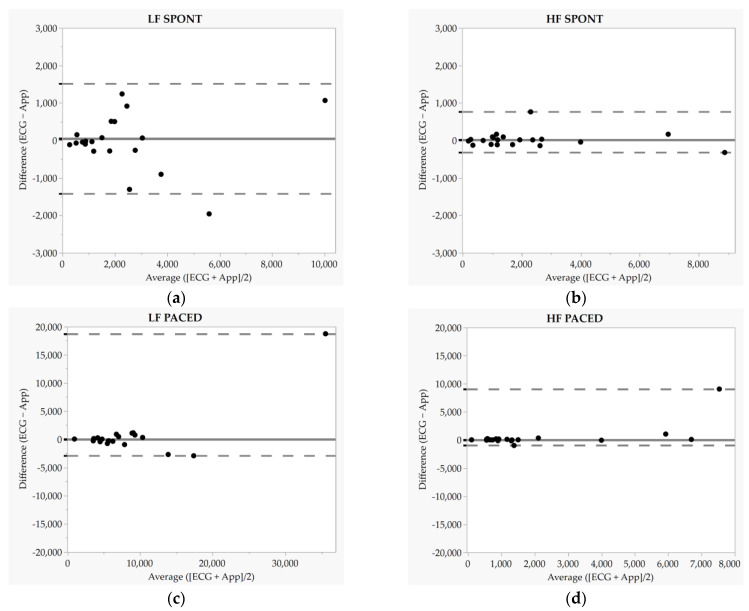
Bland–Altman plots for low-frequency (LF) and high-frequency (HF) power. Each dot represents one participant. (**a**,**b**) represent Bland-Altman plots during the spontaneous breathing condition while (**c**,**d**) represent Bland-Altman plots during the paced breathing (6 breaths·min^−1^) condition. SPONT = spontaneous breathing; PACED = paced breathing (6 breaths·min^−1^). Solid lines represent median biases, and dashed lines represent 2.5th and 97.5th percentiles. In panel a (LF SPONT), the dashed line represents mean bias and dashed lines represent 95% limits of agreement.

**Table 1 sensors-23-09496-t001:** Beat correction by participant.

Participant	ECG SPONT	ECG PACED	App SPONT	Signal Quality	App PACED	Signal Quality
1	4 ^$^	4 ^$^	0	good	0	good
2	0	2 ^$^	0	good	0	good
3	0	0	0	good	0	good
4	0	0	0	good	0	good
5	0	0	0	good	2	good
6	0	0	6	good	46	okay
7	0	0	0	good	0	good
8	0	0	0	good	0	good
9	0	0	0	good	0	good
10	0	0	0	good	0	good
11	0	0	0	good	0	good
12	0	0	0	good	0	good
13	0	0	0	good	0	good
14	0	0	0	good	0	good
15	0	0	0	good	0	good
16	0	0	0	good	0	good
17	0	0	2	good	2	okay
18	0	0	0	good	0	good
19	0	0	0	good	0	good
20	2 ^‡^	4 ^‡^	2	good	6	good

ECG = electrocardiogram; App = Elite HRV smartphone application; SPONT = spontaneous breathing; PACED = paced breathing (6 breaths·min^−1^); ^$^ = ectopic beat correction; ^‡^ = R-peak identification error correction

**Table 2 sensors-23-09496-t002:** Device comparison summary and agreement statistics: time domain.

			Median (IQR)	*p*	Median Bias (IQR)	Limits of Agreement		τ	LCC	Ordinary Least Products Regression		
						Lower 2.5th	Upper 97.5th			R^2^	Slope (95% CI)	Intercept (95% CI)
**Mean RR** **(ms)**	SPONT	ECG	1040.8 (268.4)	0.60	−0.04 (2.3)	−19.8	29.1	0.36	1.0	0.99	1.0 (0.98–1.03)	−4.5 (−27.9–18.9)
		App	1026.2 (249.7)									
	PACED	ECG	1018.3 (207.2)	0.002	1.5 (5.3)	−1.6	114.6	0.05	1.0	0.94	1.0 (0.9–1.1)	−8.1 (−80.1–63.8)
		App	1003.0 (198.3)									
**SDNN** **(ms)**	SPONT	ECG	84.2 (43.0)	0.01 *	1.3 (3.6)	−71.9	33.4	0.21	0.91	0.64	1.2 (0.9–1.4)	−15.6 (−45.9–14.8)
		App	84.8 (35.5)									
	PACED	ECG	100.1 (50.7)	0.52	0.1 (3.5)	−9.9	88.4	0.34	0.92	0.67	1.4 (0.9–1.9)	−37.2 (−86.1–11.7)
		App	99.6 (44.7)									
**RMSSD** **(ms)**	SPONT	ECG	61.3 (47.1)	<0.001 *	1.4 (5.5)	−1.4	34.6	0.41	0.98	0.98	1.2 (1.0–1.3)	−8.5 (−16.4–−0.5)
		App	61.1 (35.7)									
	PACED	ECG	71.6 (46.9)	0.01 *	0.4 (5.7)	−12.3	128.2	0.47	0.80	0.17	1.7 (0.7–2.6)	−41.9 (−104.2–20.5)
		App	76.9 (47.9)									

Mean RR = average normal-to-normal interval; SDNN = standard deviation of normal-to-normal intervals; RMSSD = root mean square of successive differences; SPONT = spontaneous breathing; PACED = paced breathing (6 breaths·min^−1^); ECG = electrocardiogram; App = Elite HRV smartphone application; IQR = interquartile range; *p* = *p*-value for Wilcoxon Signed Rank Test; τ = Kendall’s Tau; LCC = Lin’s concordance correlation coefficient; CI = confidence interval. * Denotes a significant difference between ECG and Elite HRV for the HRV metric during the respective condition.

**Table 3 sensors-23-09496-t003:** Condition comparison summary and agreement statistics.

HRV Metric	HRV Tool	Condition	Mean ± SD or Median (IQR)	*p*	LCC
**Mean RR (ms)**	ECG	SPONT	1040.8 (268.4)	0.04 *****	0.95
		PACED	1018.3 (207.2)		
	App	SPONT	1026.2 (249.7)	0.04 *****	
		PACED	1003.0 (198.3)		
**SDNN** **(ms)**	ECG	SPONT	90.0 ± 37.3	0.006 *	0.70
		PACED	111.9 ± 45.4		
	App	SPONT	90.3 ± 31.9	0.05	
		PACED	107.2 ± 32.6		
**RMSSD (ms)**	ECG	SPONT	61.3 (47.1)	0.13	0.65
		PACED	71.6 (46.9)		
	App	SPONT	61.1 (35.7)	0.52	
		PACED	76.9 (47.9)		
**LF** **(ms^2^)**	ECG	SPONT	1786.3 (2074.5)	<0.001 *	0.75
		PACED	6618.2 (5251.9)		
	App	SPONT	1634.0 (2084.7)	<0.001 *****	
		PACED	6356.2 (4081.8)		
**HF** **(ms^2^)**	ECG	SPONT	1321.8 (1705.2)	0.18	0.39
		PACED	965.2 (1413.0)		
	App	SPONT	1276.5 (1618.7)	0.06	
		PACED	1047.4 (1313.2)		

Mean RR = average normal-to-normal interval; SDNN = standard deviation of normal-to-normal intervals; RMSSD = root mean square of successive differences; LF = low-frequency spectral power; HF = high-frequency spectral power; ECG = electrocardiogram; App = Elite HRV app; SPONT = spontaneous breathing; PACED = paced breathing (6 breaths·min^−1^); SD = standard deviation; *p* = *p*-value for parametric or non-parametric paired *t*-test; LCC = Lin’s concordance correlation coefficient. * Denotes a significant difference between SPONT and PACED.

**Table 4 sensors-23-09496-t004:** Device comparison summary and agreement statistics: frequency domain.

			Mean ± SD or Median (IQR)	*p*	Mean Bias ± SD or Median Bias (IQR)	Limits of Agreement (95% OR 2.5th and 97.5th)		τ	LCC	Ordinary Least Products Regression		
						Lower	Upper			R^2^	Slope (95% CI)	Intercept (95% CI)
**LF** **(ms^2^)**	SPONT	ECG	2273.4 ± 2259.4	0.80	44.0 ± 748.9	−1424.0	1511.9	0.46	0.95	0.89	1.0 (0.7–1.3)	−56.9 (−578.4–464.6)
		App	2317.4 ± 2246.9									
	PACED	ECG	6618.2 (5251.9)	0.93	28.9 (1100.2)	−2901.4	18,752.9	0.67	0.85	0.63	1.6 (0.7–2.4)	−3869.6 (−9067.2–1327.9)
		App	6356.2 (4081.8)									
**HF** **(ms^2^)**	SPONT	ECG	1321.8 (1705.2)	0.96	14.2 (199.1)	−323.4	762.1	0.21	1.0	0.99	1.0 (0.9–1.1)	52.9 (−60.4–166.2)
		App	1276.5 (1618.7)									
	PACED	ECG	965.2 (1413.0)	0.13	35.1 (208.4)	−994.8	9046.4	0.32	0.66	0.49	1.7 (0.0–3.4)	−707.2 (−2177.6–763.2)
		App	1047.4 (1313.2)									

LF = low-frequency spectral power; HF = high-frequency spectral power; SPONT = spontaneous breathing; PACED = paced breathing (6 breaths·min^−1^); ECG = electrocardiogram; App = Elite HRV app; SD = standard deviation; IQR = interquartile range; *p* = *p*-value for paired *t*-test or Wilcoxon Signed Rank Test; τ = Kendall’s Tau; LCC = Lin’s concordance correlation coefficient; CI = confidence interval.

## Data Availability

Data are available upon request from the corresponding author.

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
