# Peer review of "Validity and Efficacy of the Elite HRV Smartphone Application during Slow-Paced Breathing"

_sensors, 2023, doi:10.3390/s23239496_

Round 1
Reviewer 1 Report
Comments and Suggestions for Authors
The study by Vondrasek and collaborators is interesting, however it is incipient or even considered as preliminary results. The objective is adequate. However, the methodology is lacking, especially when considering the number of investigated volunteers. 22 volunteers is not enough if we consider that women and men have a different modulatory pattern. I suggest recalculating the number of volunteers in order to achieve more robust results. I also suggest centralizing the age of the volunteers in just one decade (20 to 30 years), as well as investigating only eutrophic individuals. Another important point is that the authors often referred to autonomic modulation as a synonym for autonomic activity, which is not correct.
Finally, the manuscript as it is presented does not allow to achieve the proposed objectives. I suggest a greater investment in the methodology.
Comments on the Quality of English LanguageNo comments.
Author Response
We appreciate you taking the time to read over our manuscript. We agree that our sample size could be seen as a minor weakness, so we added this point to the limitation section. However, our sample size is similar to the sample size of other validation studies. Example: https://www.mdpi.com/1424-8220/22/24/9883.
Reviewer 2 Report
Comments and Suggestions for Authors
Abstract – “criterion electrocardiography” – how about “reference electrocardiography”?
Line 36 – you have “and” twice in a sentence – remove the first
Line 37 – need a reference.
Line 47 – too many “and” used
Line 59 – criterion – should use reference or gold standard.
Line 75 – “a convenience sample” – please rephrase and define
Line 80 – include IRB number as per Journal guidelines (yes, it’s in the end section too).
Line 86 – Was caffeine excluded? What about recent high intensity exercise, illness?
Line 101 – any conductive gel used for the Polar?
Line 104 – Please clarify the ECG recording chain, perhaps with a figure. From the text it sounds as if the Biopac software converted the ECG data into RRs, then those RRs were imported into Kubios. If this is correct, why didn’t you directly import the Biopac raw directly into Kubios? This would have facilitated identification of waveform quality, noise, arrythmia and taken advantage of Kubios R peak detection with upsampling. The Biopac RR conversion algorithm (if used) would have added another layer of error.
Line 107 – What version of Kubios (i.e., 4.0?), what preprocessing setting, detrending method was used, was artifact correction mode always set on auto? Did you have a predetermined artifact acceptance cutoff?
Line 126 – device manufacturer and model?
Line 144 – treatment and assessment of artifact is not well described. According to the Elite documentation (https://help.elitehrv.com/article/102-signal-quality-raw-corrected-rrs), they report artifact % as well as “quality”. Did you record the artifact %?
Results:
Artifacts – “filtered ECG, the percentage of artifacts corrected was < 4% for 20/21 participants (range: 0%–33.0%; median: 0%) during SPONT, and < 3% for 19/21 and < 11% for 20/21 (range: 0%–29.0%; median: 0%) during PACED.” This is very confusing! Since you have an ECG, were the “artifacts” identified as noise or arrythmia? Is it possible that the artifact correction algorithm (Kubios auto method) falsely classified the beats as artifact since the RR variation was high during paced breathing? Since artifact correction can alter HRV metrics, this needs to be explained carefully and a table of participant #, trial and artifact correction would be helpful as a supplementary file.
Line 229 – One could argue that your Lin value of .34 is not very correlated (https://www.ncbi.nlm.nih.gov/pmc/articles/PMC6107969/). The CI also includes zero which is a bit problematic. Was there a reason you did not include regression plots (at least as a supplemental file)?
Line 443 – The Kubios “auto” method is not considered “standard” or “popular”. In fact, it is only available in the paid version of Kubios and is not used in any of the smartphone apps (to my knowledge). However, methodology is documented, and one developer has implemented it (https://www.fitnesshrv.com/2022/06/12/artifacts.html). Hence, you should make it clear that the free version of Kubios does not have that method available. The same criticism can be made for the specific method of “detrending” used by Kubios (vs whatever Elite is doing). That method is (probably) not used by other smartphone apps unless specified. See Voss (https://pubmed.ncbi.nlm.nih.gov/25822720/), in particular the section “Importance of data pre-processing and stationarity”.
Evaluating the potential issues concerning disagreement between the Polar belt vs ECG should also include a comment on the difficulty in comparing waveforms between the devices (doi:10.1109/EMBC.2019.8857954) due to lead placement as well as the issues in Pan Tompkins like algorithms to determine R peaks which impact RR timing (https://doi.org/10.3390/bioengineering7020053). Both waveform alteration/lead placement (belt vs ECG) and R peak estimation (Kubios vs Polar on device algorithm) will be factors. Add to that the differences in preprocessing and artifact correction, it’s no wonder the agreement is influenced by intervals of high RR variability.
Comments on the Quality of English LanguageSee above
Author Response
Abstract – “criterion electrocardiography” – how about “reference electrocardiography”?
Thank you for the suggestion. We made this change to improve clarity.
Line 36 – you have “and” twice in a sentence – remove the first
Thank you for this suggestion. We made this change to improve the clarity of the sentence.
Line 37 – need a reference.
Thank you highlighting this gap in supporting evidence. We added references to support the statement in the paper.
Line 47 – too many “and” used
We made this change to improve clarity. Thank you for this suggestion.
Line 59 – criterion – should use reference or gold standard.
We made this change to add clarity. Thank you for this suggestion.
Line 75 – “a convenience sample” – please rephrase and define
We added a detail that the sample was recruited from the college campus to make it clearer where the convenience sample was drawn from.
Line 80 – include IRB number as per Journal guidelines (yes, it’s in the end section too).
We added this per you recommendation and the guideline of the journal. Thank you.
Line 86 – Was caffeine excluded? What about recent high intensity exercise, illness?
They were indeed both accounted for and we added details of this in the paper to improve clarity. Thank you for the suggestion.
Line 101 – any conductive gel used for the Polar?
The Polar H10 was wetted for improved conductivity and we included this detail in the methods for increased transparency of the methods.
Line 104 – Please clarify the ECG recording chain, perhaps with a figure. From the text it sounds as if the Biopac software converted the ECG data into RRs, then those RRs were imported into Kubios. If this is correct, why didn’t you directly import the Biopac raw directly into Kubios? This would have facilitated identification of waveform quality, noise, arrythmia and taken advantage of Kubios R peak detection with upsampling. The Biopac RR conversion algorithm (if used) would have added another layer of error.
We added details in the methods to make it clearer that raw files from AcqKnowledge were exported into Kubios without any filtering. Since we added these details, we did not feel that it was necessary to add a figure. Thank you for asking about this because it has improved the clarity of the Methods section.
Line 107 – What version of Kubios (i.e., 4.0?), what preprocessing setting, detrending method was used, was artifact correction mode always set on auto? Did you have a predetermined artifact acceptance cutoff?
We have added the version for Kubios within the text. In light of this comment, and your related comments below, we completely revised our HRV analysis methods. All reference HRV values have now been computed via Kubios following manual inspection and correction for ectopic beats. We have added details of this methodology within the paper. In this revised approach, we did not apply any detrending methods since Elite HRV makes no mention of detrending in their resources.
Line 126 – device manufacturer and model?
This information was added. Thank you for this suggestion.
Line 144 – treatment and assessment of artifact is not well described. According to the Elite documentation (https://help.elitehrv.com/article/102-signal-quality-raw-corrected-rrs), they report artifact % as well as “quality”. Did you record the artifact %?
We added a table which includes the number of artifacts corrected in ECG processing or automatically by the app for improved clarity.
Results:
Artifacts – “filtered ECG, the percentage of artifacts corrected was < 4% for 20/21 participants (range: 0%–33.0%; median: 0%) during SPONT, and < 3% for 19/21 and < 11% for 20/21 (range: 0%–29.0%; median: 0%) during PACED.” This is very confusing! Since you have an ECG, were the “artifacts” identified as noise or arrythmia? Is it possible that the artifact correction algorithm (Kubios auto method) falsely classified the beats as artifact since the RR variation was high during paced breathing? Since artifact correction can alter HRV metrics, this needs to be explained carefully and a table of participant #, trial and artifact correction would be helpful as a supplementary file.
We appreciate your feedback on this. We have revised this section and included a new table with the number of artifacts that were corrected for each participant of the study.
Line 229 – One could argue that your Lin value of .34 is not very correlated (https://www.ncbi.nlm.nih.gov/pmc/articles/PMC6107969/). The CI also includes zero which is a bit problematic. Was there a reason you did not include regression plots (at least as a supplemental file)?
Since we reperformed the analysis without using the automatic filter and without detrending, this comment is no longer relevant. The response to the next comment will help provide context.
Line 443 – The Kubios “auto” method is not considered “standard” or “popular”. In fact, it is only available in the paid version of Kubios and is not used in any of the smartphone apps (to my knowledge). However, methodology is documented, and one developer has implemented it (https://www.fitnesshrv.com/2022/06/12/artifacts.html). Hence, you should make it clear that the free version of Kubios does not have that method available. The same criticism can be made for the specific method of “detrending” used by Kubios (vs whatever Elite is doing). That method is (probably) not used by other smartphone apps unless specified. See Voss (https://pubmed.ncbi.nlm.nih.gov/25822720/), in particular the section “Importance of data pre-processing and stationarity”.
We greatly appreciate this comment because it led to the complete reevaluation of our results. Now that we have redone the analysis, we have a stronger rationale for what we did. Instead of applying the autocorrection filter, we manually inspected each ECG file and manually corrected as appropriate. We also did not use detrending (optional in Kubious). Full details are now included in the manuscript.
Evaluating the potential issues concerning disagreement between the Polar belt vs ECG should also include a comment on the difficulty in comparing waveforms between the devices (doi:10.1109/EMBC.2019.8857954) due to lead placement as well as the issues in Pan Tompkins like algorithms to determine R peaks which impact RR timing (https://doi.org/10.3390/bioengineering7020053). Both waveform alteration/lead placement (belt vs ECG) and R peak estimation (Kubios vs Polar on device algorithm) will be factors. Add to that the differences in preprocessing and artifact correction, it’s no wonder the agreement is influenced by intervals of high RR variability.
Thanks you for this helpful comment. We have substantially elaborated on the potential sources of error in the Discussion with your recommendations along with other possibilities.
Reviewer 3 Report
Comments and Suggestions for Authors
1. In lines 11 to 14 (Abstract), the authors mention that “20 healthy young adults (13 M/8 F) completed one counterbalanced cross-over protocol involving 10 min each of supine spontaneous (SPONT) and paced (PACED ; 6 breaths•min-1 ) breathing while RR intervals were simultaneously recorded via a Polar H10 paired with Elite HRV and criterion electrocardiography (ECG)”. Here, the sum of 13M+8F would indicate a total of 21 patients and not 20. It needs to be changed to the correct value.
2. In line 75, Material and Methods, there is again an error in the information about the number of patients studied. It is mentioned that “We recruited a convenience sample of 22 healthy adults (13 M/8 F)”. The sum of 13M+8F is 21 and not 22 as mentioned. Need to change to correct value.
3. The P values reported in the text (as in the RMSSD analysis) do not coincide with those reported in Table 1. It also happened with other variables. This greatly compromises the interpretation of the results and needs to be reviewed.
4. Another consideration: Some measurements, such as the RMSSD Paced App, show values that are not compatible with clinical reality. With an RMSSD of 142.6 ± 178.5 ms, this would include individuals with an RMSSD of near 0 to 220 ms, which is not compatible with the reality in healthy young individuals. For the SDNN Paced App, the value of 136.1 ± 115.4 ms is reported, which also suggests measurements with artifact interference. Perhaps the controlled breathing mode was not properly assimilated by the participants, causing changes in the records. Need revision
Comments on the Quality of English Language
Nothing to comment.
Author Response
Reviewer #3
- In lines 11 to 14 (Abstract), the authors mention that “20 healthy young adults (13 M/8 F) completed one counterbalanced cross-over protocol involving 10 min each of supine spontaneous (SPONT) and paced (PACED ; 6 breaths•min-1 ) breathing while RR intervals were simultaneously recorded via a Polar H10 paired with Elite HRV and criterion electrocardiography (ECG)”. Here, the sum of 13M+8F would indicate a total of 21 patients and not 20. It needs to be changed to the correct value.
We update the analysis and there are now 20 participants and we made sure that this is consistent across the manuscript. Thank you.
- In line 75, Material and Methods, there is again an error in the information about the number of patients studied. It is mentioned that “We recruited a convenience sample of 22 healthy adults (13 M/8 F)”. The sum of 13M+8F is 21 and not 22 as mentioned. Need to change to correct value.
We update the analysis and there are now 20 participants and we made sure that this is consistent across the manuscript. Thank you.
- The P values reported in the text (as in the RMSSD analysis) do not coincide with those reported in Table 1. It also happened with other variables. This greatly compromises the interpretation of the results and needs to be reviewed.
We updated the results and made sure that this error was not repeated. Thank you for this suggestion.
- Another consideration: Some measurements, such as the RMSSD Paced App, show values that are not compatible with clinical reality. With an RMSSD of 142.6 ± 178.5 ms, this would include individuals with an RMSSD of near 0 to 220 ms, which is not compatible with the reality in healthy young individuals. For the SDNN Paced App, the value of 136.1 ± 115.4 ms is reported, which also suggests measurements with artifact interference. Perhaps the controlled breathing mode was not properly assimilated by the participants, causing changes in the records. Need revision
We updated the results and removed the unfiltered data which makes this comment no longer relevant, but we appreciate your insight.
Reviewer 4 Report
Comments and Suggestions for Authors
The authors present results from a study in which they tested a smartphone application to measure HRV parameters at rest and during a slow-paced breathing condition. Overall, this is a good manuscript, with a clear introduction, methodology, and discussion, meaning I only have a few minor comments/suggestions.
Introduction, line 53: Before the presentation of the used smartphone application, I think that it could be interesting to add information on other validated and tested smartphone applications that were used to measure HRV parameters (e.g., HRV4Training app, Heart Rate Variability HRV Camera app, etc.). This increases the completeness of the study background, showing that the study follows a research line that tries to increase the reliability of mHealth devices.
Materials and Methods, line 92: If available, more information on how the smartphone application automatically detects/manages artefacts and computes the HRV parameters should be added.
Materials and Methods, lines 115-123: The rationale for choosing the used HRV parameters is clear, but why did the authors not use other interesting parameters such as pNN50 or LF/HF? Are these not provided by the application?
Materials and Methods, Statistical Analysis: Considering the small sample and the typical non-normal distribution of HRV variables, I hypothesised that data have undergone a logarithmic transformation. Is it correct, or were all data normally distributed?
Results: Were all the included participants compliant with the app-guided slow-paced breathing procedure?
Results, line 173: I did not understand if the analysed sample was 21 (as was written here) or 20 (as was written in the abstract).
Discussion: At the end of the discussion, I would like to add a brief section in which the authors explain these findings' practical and clinical value, focusing on the added value for mHealth research.
Author Response
The authors present results from a study in which they tested a smartphone application to measure HRV parameters at rest and during a slow-paced breathing condition. Overall, this is a good manuscript, with a clear introduction, methodology, and discussion, meaning I only have a few minor comments/suggestions.
We appreciate your compliment and the time you took to review our manuscript.
Introduction, line 53: Before the presentation of the used smartphone application, I think that it could be interesting to add information on other validated and tested smartphone applications that were used to measure HRV parameters (e.g., HRV4Training app, Heart Rate Variability HRV Camera app, etc.). This increases the completeness of the study background, showing that the study follows a research line that tries to increase the reliability of mHealth devices.
We appreciate this suggestion. Respectfully, we did not make this revision because the focus of the paper is specifically on tools for the practice of slow-paced breathing or HRV biofeedback.
Materials and Methods, line 92: If available, more information on how the smartphone application automatically detects/manages artefacts and computes the HRV parameters should be added.
Thank you for this comment. Unfortunately, Elite HRV is not transparent with their RR filtering method. We have now mentioned in the text that this is a proprietary process.
Materials and Methods, lines 115-123: The rationale for choosing the used HRV parameters is clear, but why did the authors not use other interesting parameters such as pNN50 or LF/HF? Are these not provided by the application?
We did not include pNN50 because it is rarely used in the field for HRV tracking, whether it be for biofeedback or daily ANS assessment. We did not include the LF/HF ratio because it has been heavily criticized and its physiological meaning is somewhat ambiguous. For example:
https://www.frontiersin.org/articles/10.3389/fphys.2013.00026/full
Materials and Methods, Statistical Analysis: Considering the small sample and the typical non-normal distribution of HRV variables, I hypothesised that data have undergone a logarithmic transformation. Is it correct, or were all data normally distributed?
Thank you for this helpful observation. We have completely revised our statistical approach to account for the non-normal distribution of difference scores. We did not use Ln transformed variables because this method, by design, limits outliers, which is unhelpful for individuals interested in raw values. However, we now compare our findings of raw values with previous studies that used Ln values, of which tended to show better agreement.
Results: Were all the included participants compliant with the app-guided slow-paced breathing procedure?
Yes, based on the breathing rate for the PACED breathing condition, all participants were compliant. A breathing rate of 6 breaths per minute was confirmed in the results section.
Results, line 173: I did not understand if the analysed sample was 21 (as was written here) or 20 (as was written in the abstract).
We updated the result and there are 20 people included in the final analysis. We made sure that this was consistently reported throughout the manuscript.
Discussion: At the end of the discussion, I would like to add a brief section in which the authors explain these findings' practical and clinical value, focusing on the added value for mHealth research.
We agree that a more practical summation of the findings for practitioners is warranted. We have completely revised the conclusion in light of the new analyses. I hope our practical conclusion satisfies your concern.
Reviewer 5 Report
Comments and Suggestions for Authors The submitted manuscript presents data from a study aiming to assess the efficacy and validity of a smartphone app for assessment of heart rate variability (HRV) during slow-paced breathing. It should be noted that the standard measurement parameters for HRV have not changed since 1996, when the only European-American guideline was issued. Since then, all publications seem to have been divided into two categories of technical and medical focus. The first are devoted to new technologies for measuring HRV, while medical publications report on the significance of HRV in various clinical settings. The present article seems to be trying to close this existing gap. It cannot be clearly classified into either of the two categories. In general, the study was clearly conducted, presented in a well-structured manner, and provides relevant results. The performed statistical analysis is convincing. I have no significant remarks. The manuscript may be recommended for publication.Author Response
We appreciate you taking the time to review this paper.
Round 2
Reviewer 2 Report
Comments and Suggestions for Authors
Thank you for the redo - looks great - congratulations.
Comments on the Quality of English LanguageNone